# Identification of Crucial Residues in α-Conotoxin EI Inhibiting Muscle Nicotinic Acetylcholine Receptor

**DOI:** 10.3390/toxins11100603

**Published:** 2019-10-16

**Authors:** Jiong Ning, Jie Ren, Yang Xiong, Yong Wu, Manqi Zhangsun, Dongting Zhangsun, Xiaopeng Zhu, Sulan Luo

**Affiliations:** Key Laboratory of Tropical Biological Resources, Ministry of Education, Key Lab for Marine Drugs of Haikou, School of Life and Pharmaceutical Sciences, Hainan University, Haikou 570228, China; ningjiong2018@163.com (J.N.); hndx2303@163.com (J.R.); ncdxxy@163.com (Y.X.); wuyong@hainu.edu.cn (Y.W.); zhangsunmanqi@163.com (M.Z.); zhangsundt@163.com (D.Z.)

**Keywords:** Muscle-type nAChR, α-CTx EI, Ala-scan mutagenesis, TEVC, CD spectroscopy

## Abstract

α-Conotoxins (α-CTxs) are small disulfide-rich peptides from venom of *Conus* species that target nicotinic acetylcholine receptors (nAChRs). The muscle-type nAChRs have been recognized as a potential target for several diseases, such as myogenic disorders, muscle dystrophies, and myasthenia gravis. EI, an α4/7-CTx, mainly blocks α1β1δε nAChRs and has an extra N-terminal extension of three amino acids. In this study, the alanine scanning (Ala-scan) mutagenesis was applied in order to identify key residues of EI for binding with mouse α1β1δε nAChR. The Ala-substituted analogues were tested for their abilities of modulating muscle and neuronal nAChRs in *Xenopus laevis* oocytes using two-electrode voltage clamp (TEVC) recordings. Electrophysiological results indicated that the vital residues for functional activity of EI were His-7, Pro-8, Met-12, and Pro-15. These changes exhibited a significant decrease in potency of EI against mouse α1β1δε nAChR. Interestingly, replacing the critical serine (Ser) at position 13 with an alanine (Ala) residue resulted in a 2-fold increase in potency at the α1β1δε nAChR, and showed loss of activity on α3β2 and α3β4 nAChRs. Selectivity and potency of [S13A] EI was improved compared with wild-type EI (WT EI). In addition, the structure–activity relationship (SAR) of EI revealed that the “Arg1–Asn2–Hyp3” residues at the N-terminus conferred potency at the muscle-type nAChRs, and the deletion analogue ^△1–3^ EI caused a total loss of activity at the α1β1δε nAChR. Circular dichroism (CD) spectroscopy studies demonstrated that activity loss of truncated analogue ^△1–3^ EI for α1β1δε nAChR is attributed to disturbance of the secondary structure. In this report, an Ala-scan mutagenesis strategy is presented to identify crucial residues that are significantly affecting potency of E1 for mouse α1β1δε nAChR. It may also be important in remodeling of some novel ligands for inhibiting muscle-type nAChRs.

## 1. Introduction

Nicotinic acetylcholine receptors (nAChRs) are composed of five subunits that are arranged around a central cation pore, and they are a member of the ligand-gated ion channel superfamily [1,2,3]. The nAChRs are further classified into muscular and neuronal subtypes. The muscle-type receptors are mainly distributed at the skeletal neuromuscular junction and form heteropentamers composed of two α1, one β1, one δ, and one γ/ε subunit [4,5]. Previous investigations demonstrated that muscle-type nAChRs are associated with pathophysiology conditions, including myogenic disorders, muscle dystrophies, and myasthenia gravis [6,7]. Several toxins that act specifically on muscle-type nAChR have entered the preclinical applications, including Azemiopsin (Az), a neuropeptide from the *Azemiops feae* viper venom. They are a high selectivity antagonist of muscle-type nAChR and are regarded with high potential for application to nondepolarizing muscle relaxants [8]. Another α-bungarotoxin TFT, was discovered almost 50 years ago and has been widely used as a specific antagonist for neuro- and muscle-type nAChRs [9].

α-Conotoxins (α-CTxs) show high affinities with muscle nAChRs and are small, disulfide-rich peptide toxins isolated from the venom of predatory marine snails (genus *Conus*), ranging typically from 12 to 20 amino acids in size [10,11]. These toxins block muscle-type and neuronal-type nAChRs with high potency and selectivity; for example, GIC selectively inhibited α3β2 nAChR with IC_50_ of 1.1 nM [11]. α-CTxs have been used as valuable tools in understanding the mechanism involved in ligand–receptor interaction for nearly 30 years [12,13]. α-CTx EI is an 18-residue conotoxin, with a 4/7 intercysteine framework, isolated from the venom of *Conus ermineus* [14]. The sequence of EI is shown in Table 1. Unlike other neuronal selective α-CTxs, EI has a three amino acid N-terminal tail and a post-modification hydroxyproline (O) residue. EI potently inhibited α1β1δε nAChR with IC_50_ values of 187 nM and is about 100-fold less potent at α3β2 and α3β4 combinations [15,16]. Compared with other α4/7-CTxs, EI has an extra N-terminal tail and preference for muscle-type nAChRs versus neuronal-type nAChRs.

Previous studies have shown that the structure of EI was resolved by NMR and X-ray diffraction methods [16,17]. However, the role of each residue in α-CTx EI remains partially unknown. The Ala-scan mutagenesis is a widely-used approach for exploring the structure–activity relationship (SAR) between receptors and ligands, and identifying key positions in protein that are important for function or ligand affinity [18]. In this report, EI and its analogues were synthesized and characterized (Table 1). Electrophysiological results indicate an N-terminal tail and a Ser-13 in α-CTx EI that contribute to α1β1δε nAChR potency and selectivity. A Pro in position 15 has obvious effects on potency of EI. Substitution of Ala for Pro-15 resulted in complete loss of activity compared with wild-type (WT) EI at both α1β1δε nAChR and other neuronal nAChRs. In addition, the circular dichroism (CD) spectroscopy method was applied to further explore the secondary structure of EI and its analogues. The results of these studies provide valuable insight into the designing ligand that selectively targets muscle nAChRs.

## 2. Results

### 2.1. Peptide Synthesis and Oxidative Folding of α-Conotoxin EI and its Analogues

For α-CTx EI and its analogues, these linear peptides were successfully synthesized using standard solid phase peptide synthesis with Fmoc (9-fluorenylmethoxycarbonyl) chemistry. α-CTxs have four cysteine residues with an CC–Xm–C–Xn–C framework that yields three possible disulfide bond connectivities: globular (I–III, II–IV), ribbon (I–IV, II–III), and beads (I–II, III–IV). Typically, the globular conformation occurred in natural α-CTxs [19]. Acm-protected Cys residues were therefore incorporated at positions 2 and 4 in α-CTx EI and analogues. The two-step oxidation is a widely-used approach for folding these peptides into the corresponding disulfide conformation. Briefly, the formation of first disulfide bond in each peptide incubated in 5 mM ferricyanide (pH 7.5, 2 mg/mL) 45 min at 25 °C. The monocyclic peptide was purified by preparative RP-HPLC. Closure of the second disulfide bond was treated with 0.4 mM I_2_ (0.4 mg/mL) containing 1% TFA under nitrogen protection conditions for 10 min. Then 1 M ascorbic acid was gently added to terminate the reaction. The bicyclic product was again purified by preparative RP-HPLC. The fully oxidized peptide EI was detected by analytical RP-HPLC with the retention time of 12.31 min (Figure 1A), and ESI–MS of synthetic α-CTx EI had an observed monoisotopic mass of 2092.82 Da, which was consistent with the theoretical molecular weight (2092.84 Da) (Figure 1B). Similarly, the observed molecular mass (2076.80 Da) of [S13A] EI was consistent with the theoretical molecular weight (2076.82 Da) (Figure 1C,D). A series of EI analogues were synthesized following the same protocol, and their HPLC chromatogram and mass spectrometry (MS) profiles are provided in the Appendix A.

### 2.2. Ala-SCAN of the Inter-Cysteine Loops Revealed Key Residues for EI Activity

To better understand the SAR between α-CTx EI and nAChRs, the Ala-substituted analogues were tested for the ability to modulate muscle and neuronal nAChRs in *Xenopus laevis* oocytes using two-electrode voltage clamp (TEVC) recordings. Figure 2 shows the relative amount of inhibition EI analogues generated comparing with WT EI at different concentrations. We observed that EI exhibited no activity at α1β1δε nAChR at the concentration of 10 nM, and it displayed only weak inhibition against muscle nAChR by incubation with 100 nM EI (Figure 2A (I)). In contrast, with incubation of 100 nM [S13A] EI, the inhibition of α1β1δε nAChR current response to 10 μM ACh was 80.4 ± 2.5% (*n* = 3) (Figure 2A (II)). Concentration–response curves of EI and [S13A] EI were subsequently assessed on mouse α1β1δε nAChR. Figure 3C and Table 2 revealed that [S13A] EI inhibited α1β1δε nAChR with IC_50_ of 34.80 nM, a 2-fold higher potency than WT EI. High concentrations of [S13A] EI and EI were also tested at the α3β2 and α3β4 nAChRs; 1 μM EI produced 52.5 ± 3.2% (*n* = 3) and 48 ± 4.5% (*n* = 3) inhibition at the α3β2 and α3β4 nAChRs, respectively (Figure 2B (I) and 2C (I)). However, [S13A] EI showed different potency for both α3β2 and α3β4 nAChR subtypes. With incubation of 1 μM [S13A] EI, the inhibition of α3β2 and α3β4 nAChRs current response to 10 μM ACh was 9.5 ± 2.0% (*n* = 3) and 3 ± 0.8% (*n* = 3), respectively (Figure 2B (II) and 2C (II)). In conclusion, the selectivity of [S13A] EI was vastly improved at muscle and neuronal nAChRs versus WT EI. Replacing residue Tyr-6 with Ala or the substitution of Thr-9 with Ala all caused a decrease in potency at the α1β1δε nAChR but substantially exhibited an increase of potency for α3β2 and α3β4 nAChR. [T9A] EI displayed an 11-fold and a 112-fold increase in activity at α3β4 and α3β2 nAChR, respectively, compared with WT EI (Table 2). Most Ala mutants in α-CTx EI exhibited a moderate decrease in antagonist activity for mouse α1β1δε nAChR, including [R1A] EI, [N11A] EI, and [N14A] EI (Figure 3A,C). Figure 3 and Table 2 reveal that replacement of His-7, Pro-8, Met-12, or Pro-15 with Ala resulted in significant reductions in activity (Figure 3B,C), especially, [P15A] EI displayed a 190-fold decrease in potency at mouse α1β1δε nAChR than native EI (Figure 3C). Four other mutants, [D2A] EI, [O3A] EI, [Q16A] EI, and [I17A] EI, preserved similar potency compared with WT EI (Figure 3A,C).

### 2.3. N-Terminal Amino Acids in EI Influence the Activity of Peptide for α1β1δε nAChR

To measure the effect of N-terminal amino acids for mouse α1β1δε nAChR, three N-truncated analogues were designed based on the number of N-terminal amino acids. Noticeably, two EI analogues, ^△1–2^ EI and ^△1–3^ EI, had a significant impact on potency for mouse α1β1δε nAChR. Figure 4 reveals that ^△1–2^ EI and ^△1–3^ EI exhibited little effect on α1β1δε nAChR at the concentration of 1 μM. These two analogues displayed no activity at other neuronal nAChRs including α3β2 and α3β4 nAChRs, even at the concentrations up to 10 μM. Subsequently, the concentration–response curves for all three N-truncated analogues are shown in the Figure 3D and the IC_50_ values of truncations at the N-terminus are summarized in Table 2. ^△1^ EI inhibited the α1β1δε nAChR with the IC_50_ of 716.4 nM and the potency of ^△1^ EI at the α1β1δε nAChR was 11-fold less than native EI. The activities of the other N-truncated analogues, including ^△1–2^ EI and ^△1–3^ EI, whose potency for α1β1δε nAChR were 180-fold lower than WT EI, and had an abolishment in α3β2 as well as α3β4 nAChRs. Above all, the fact demonstrated that the triple amino acids “RDO” at the N-terminus maintained crucially the potency of EI for mouse α1β1δε nAChR.

### 2.4. Circular Dichroism Analysis

CD spectra were obtained for aqueous solutions of native EI and their Ala-substituted analogues [P8A] EI, [S13A] EI, and [P15A] EI, as well as N-terminally truncated analogue ^△1–3^ EI. Similar overall spectra were overlaid for [S13A] EI and EI with a positive ellipticity at 195 nm (λ), and two negative ellipticities at 208 nm (λ) and 222 nm (λ), respectively (Figure 5A). This indicated that these peptides were predominantly indicative of α-helical. Replacement of Pro-8 or Pro-15 with Ala resulted in a remarkable change in spectral characteristic, and the other truncated analogue ^△1–3^ EI gave CD spectra indicative of random coil structure as well (Figure 5B).

## 3. Discussion

So far, several toxins have been reported to block muscle-type nAChRs, and α-CTxs are the most studied and pharmacologically characterized. It has been previously shown that α-CTxs isolated from fish hunting cone snail venoms target mammalian and/or fish neuromuscular nAChRs, which generally consist of 3/5 framework; while α-CTxs with a 4/7 loop motif primarily target mammalian neuronal nAChRs. However, α-CTx EI is an exception, exhibiting muscle-type nAChR [11]. The α3/5-CTx GI, MI, and SI are amongst the first nicotinic antagonists from the cone snail venoms. They are antagonist of muscle type nAChRs with a high selectivity for muscle versus neuronal subtype nAChRs [20,21]. α4/7-CTxs are widely used as nAChR antagonists, which are found in the venom of cone snails [22]. Table 3 summarizes some α4/7-CTxs that inhibited a variety of nAChR subtypes in the last fifteen years. For example, α4/7-CTx RegIIA, which was isolated from the venom of *Conus lividus,* is an antagonist of α3β2 nAChR and it is also active at α3β4 and α7 nAChRs, but has no activity at muscle-type nAChRs [23]. Another example, α4/7-CTx GID, which was from *Conus geographus*, was isolated from crude venom using RP-HPLC. GID inhibits α7 and α3β2 nAChRs with nM affinity and exhibits at least 1000-fold less potency at muscle nAChR. Unlike other α4/7-CTxs, GID has N-terminal tail of four amino acids, two post-translationally modified residues, but lacks amidated C-terminus [24]. α4/7-CTx EI is an exception, and preferentially targets α/δ subunit interface versus α/γ in mammalian muscle nAChR [14]. In this study, a series of single point mutants of α-CTx EI were synthesized and their potency was identified for α1β1δε nAChR expressing in the *Xenopus laevis* oocytes. These analogues were also screened at other neuronal nAChR subtypes, such as α3β2 and α3β4. The identification of key residues in EI is vitally important to clarify the interaction mechanism with muscle-type, α3β2, and α3β4 nAChRs binding sites.

In this report, Ala-scan mutagenesis of α4/7-CTx EI was applied to examine the interaction between EI and α1β1δε nAChR. Substitution of His-7, Pro-8, Met-12, and Pro-15 significantly reduced the potency of EI on α1β1δε nAChR. Ala substitution of Arg-1, Asn-11, and Asn-14 led to a more than a 3-fold loss of activity at the α1β1δε nAChR versus WT EI. Four mutants, [D2A] EI, [O3A] EI, [Q16A] EI, and [I17A] EI maintained the potency for α1β1δε nAChR. Remarkably, substitution of residue Ser-13 with Ala increased the potency for α1β1δε nAChR but nearly abolished activity for neuronal nAChR subtypes, and activity of [S13A] EI increased 2-fold at the α1β1δε nAChR relative to native EI.

A conserved SHPA motif in loop1 region in α4/7-CTxs is common and responsible for receptor binding activity [11,12]. The amino acids His and Pro in α-CTxs loop1 are commonly thought to confer rigidity and stability to the α-helical structure, and Pro in loop1 substituted by Ala completely abolished the activity of the main receptor. Previously study of α-CTx GID, Millard et al. [27], revealed that replacing Pro in this region with Ala resulted in a total loss of the activity for α4β2 nAChR. Similarly, Hone et al. [28], demonstrated that α-CTx [P6A] PeIA mutant exhibited approximately 580-fold lower activity of α3β2 nAChR versus PeIA. However, in this study, when Pro was substituted with Ala in loop1, its potency for α1β1δε nAChR lowered 7-fold relative to native EI. It was found that substitution of Tyr-6 and Thr-9 with Ala, respectively, had little effect on potency for α1β1δε nAChR. Interestingly, the activity of [Y6A] EI and [T9A] EI dramatically increased at the neuronal nAChRs compared with WT EI, [T9A] EI mutant exhibited more than 10-fold potency for α3β4 nAChR, and the activity of α3β2 nAChR increased greater than 100-fold compared with native EI. The reason why activity of [T9A] EI shifted from muscle nAChR to neuronal nAChRs is the difference between receptor subunits in extracellular region. The mechanism between [T9A] EI and α3β2 requires further elucidation.

It has been previously established that structural determinants in the first intracellular loop1 is primarily responsible for binding, and that a subset of residues in loop2 is of vital importance in subtype selectivity [29,30]. α-CTx TxID, Wu et al. [31], revealed that Ser in the 9^th^ position substituted with Ala caused a 46-fold loss in potency for α6β4 nAChR and maintained the activity of α3β4 nAChR, and selectivity of [S9A] TxID between α6β4 and α3β4 nAChR was improved versus native TxID. In this work, replacing residue Ser in loop2 led to little change of the potency for neuronal nAChRs but increased potency for α1β1δε nAChR by 2-fold. Consequently, we can deduce that residues in loop2, especially Ser-13, had preferences for mouse α1β1δε nAChR.

Previous studies have shown that most α4/7-CTx mainly inhibited various neuronal nAChRs [2], whereas α4/7-CTx EI mainly blocked muscle-type nAChRs with an unusual N-terminal tails. Several studies have revealed that the tails in the N-terminus in most α-CTxs often play a pivotal role in binding receptors [12]. For instance, α-CTx PIA mainly inhibited neuronal α6β2 nAChR with an extended Arg–Asn–Pro tail at the N-terminus and had the highest sequence homology with EI (Appendix A) [25]. Three truncated analogues, ^△1^ PIA, ^△1–2^ PIA, and ^△1–3^ PIA, exhibited different affinities for α6β2 nAChR in competition binding studies [32]. Especially, ^△1–3^ PIA significantly reduced affinity and potency for α6β2 nAChR comparing with native PIA. It was demonstrated that intrinsically disordered proteins (RDP) in solution but N-terminus residues outside the cysteine framework in PIA form a stable β1 secondary structure and yield biological functions [32]. A similar investigation was also performed on four residues at the N-terminus of α-CTx GID (isoleucine, arginine, aspartic acid, and γ-carboxyglutamic acid) and deletion of the N-terminal sequence resulted in inactivity for the main receptor α4β2 nAChR [27]. These results suggest that the IRDγ region in GID can contribute to α4β2 nAChR activity (Appendix A) [24]. These results also indicate that the N-terminal extension in GID plays a crucial role in maintaining the α-CTxs folding [27]. To further explore the role of N-terminus in EI on the activity of EI at the α1β1δε nAChR, a series of N-terminal truncations were also synthesized in the α-CTx EI, and the activities of ^△1–2^ EI and ^△1–3^ EI analogues caused total loss of potency at the α1β1δε nAChR in comparison to EI ^WT^. It can be therefore concluded that the N-terminal residues, especially Hyp at the position 3 in the α-CTx EI, play an important role in activity to muscle-type nAChR.

Two groups revealed the structure of EI using NMR spectroscopy and X-ray crystallography methods [14,17]. A typical structure of EI features was composed of two α-helixes, Hyp3-Tyr6 and Pro8-Ser13, and a β-turn involving residues Asn14–Ile17. In this work, the secondary structure of native EI and its analogues were examined using circular dichroism spectroscopy and it was reported that three mutants were, [P8A] EI, [P15A] EI, and ^△1–3^ EI, which are structurally different from native EI. It therefore was deduced that a change in secondary structure caused the loss in toxin potency. However, the structure of [S13A] EI resembled native EI; the increase in toxin potency might not damage the secondary structure but the increase in binding between [S13A] EI and α1β1δε nAChR. The mechanism between [S13A] EI and muscle nAChR needs to be further elucidated.

In summary, we identified residues in EI crucial for interaction with muscle-type nAChRs. The aim was to help improve the design of EI analogues that selectively and potently target receptors involved with muscle-type nAChRs.

## 4. Materials and Methods

### 4.1. Synthesis and Purification of EI and Analogues

EI and its analogues were assembled on rink amide resin using solid-phase peptide synthesis (SPPS) with an ABI 433A peptide synthesizer (Applied Biosystem, Stafford, TX, USA) and a Fmoc (N-(9-fluorenyl) methoxycarbonyl) chemistry; the procedure was performed as previously described [33,34]. In brief, we protected the cysteine residues in pairs with either S-trityl (S-Trt) on Cys1 and Cys3, or S-acetamidomethyl (S-Acm) on Cys2 and Cys4. The resin was cleaved from peptides using reagent K (trifluoroacetic acid (TFA)/phenol/thioanisole/water/ethanedithiol; 90:7.5:5:5:2.5). The reaction solvent was evaporated and the remaining mixture in peptides was precipitated with ice-cold ether, then washed with ice-cold ether twice, finally filtered, dissolved in 60% buffer B (60% CH_3_CN in H_2_O contain 0.05% TFA), and lyophilized. The peptide mixture was purified by RP-HPLC on a Vydac C18 column using a linear gradient of Buffer B ranging from 5 to 45% over 40 min. The elution was monitored with UV detector monitored at 214 nm. Buffer B was 90% acetonitrile (ACN) and 0.050% TFA in remainder H_2_O and buffer A was 0.075% TFA in remainder H_2_O. The molecular mass of the fractions was confirmed by electrospray–mass spectroscopy (ESI–MS).

### 4.2. cRNA Preparation

Plasmid DNAs that encoded various nAChR subunits were prepared as described previously [31]. These plasmids containing gene encoding mouse muscle-type nAChR subunits and rat neuro-type nAChR subunits were linearized by digestion with restriction enzymes *Sma* I (muscle-type nAChR), *EcoR* I (rα3), *Hind* III (rβ2), and *Xho* I (rβ4). The 5′-terminal capped cRNAs were synthesized in vitro from there corresponding cDNA linearized templates using SP6, T7, and T3 mMESSAGE mMACHINE transcription Kit (Ambion, Austin, TX, USA). The cRNA was purified using MEGAclear^Tm^ Transcription Purification Kit (Ambion, Austin, TX, USA). Their concentration was confirmed by Smart Spec^TM^ plus spectrophotometer (Bio-rad, Hercules, CA, USA), with their absorbance determined at 260 and 280 nm.

### 4.3. Oocyte Isolation and Microinjection

Oocytes (Stage V–VI) were prepared from mature female *Xenopus laevis* and digested with collagenase lasting for 40–60 min to remove follicle cells. Subsequently, the oocytes were incubated at 17 °C in sterile ND96 buffer (96.0 mM NaCl, 2.0 mM KCl, 1.8 mM CaCl_2_, 1.0 mM MgCl_2_, 5 mM HEPES, pH 7.1–7.5), and supplemented with antibiotics (10 μg/mL of streptomycin, 10 μg/mL of penicillin, and 100 μg/mL of gentamicin). Oocytes of mature were injected within 24 h of harvesting and cRNAs of various subunits were injected into each oocyte at a molar ratio of 1:1. Electrophysiological recordings were performed from 1–5 days and incubated at 17 °C after cRNA microinjection.

### 4.4. Electrophysiological Recordings

ACh induced membrane currents of whole oocyte cell were recorded 2–4 days after injection by using TEVC technique with an Axon 900A amplifier (Molecular Devices, Sunnyvale, CA, USA), and the holding potential (Vm) of nAChR was clamped at −70 mV. Electrodes were pulled from borosilicate glass, and this yielded a resistance between 0.5 and 2 megaohms (MΩ) when supplementing with 3 M KCl. During recording, the oocyte chamber was a cylindrical well (~50 μL volume) and was perfused under gravity at a flow rate of ~2 mL/min with sterile ND96 solution (supplemented with 0.1 mg/mL BSA). The oocyte was subjected to 1-s ACh pulse every minute, the concentration of ACh treat with α1β1δε, α3β2, and α3β4 nAChR subtypes were 10 μM, 100 μM, and 100 μM, respectively. Once a stable baseline current was recorded, either ND96 alone or ND96 containing various concentrations of EI and its analogues were perfusion-applied in a cylindrical well for 5 min before adding the agonist ACh.

### 4.5. Data Analysis

In order to obtain a “100% control” response before a test response, we used to normalize amplitude of the average of three recording. The concentration–response curves for EI and analogues were fitted by nonlinear regression analysis, % response = 100/(1 + ([toxin]/IC_50_)^nH^), where nH is the Hill slope, and IC_50_ indicates the inhibitory concentration of the antagonist required to produce 50% inhibition of the agonist response. All data represent mean ± S.E.M. of at least three to eight independent experiments, which were statistically analyzed using Prism 6.0 software (GraphPad Software, San Diego, CA, USA).

### 4.6. Circular Dichroism Spectroscopy

CD spectra of EI and its analogues were tested on a Jasco J-815 spectropolarimeter with 10 mm path length quartz cuvette. EI analogues were dissolved in 20 mM sodium phosphate buffer (pH 7.0) with the concentration of 43 μM. The spectra were measured in the far UV region (160–260 nm) using an average of 10 scans. The experimental parameters were set to a scanning speed of 50 nm/min, response time of 1 s, sensitivity range of 100 millidegrees, and a step resolution of 1 nm, and all the experiments were conducted in the temperature range of 17–23 °C, and the flow of nitrogen was maintained at 10 mL/min for the duration of the measurements. The data were analyzed and processed using the Jasco system software.

## Figures and Tables

**Figure 1 toxins-11-00603-f001:**
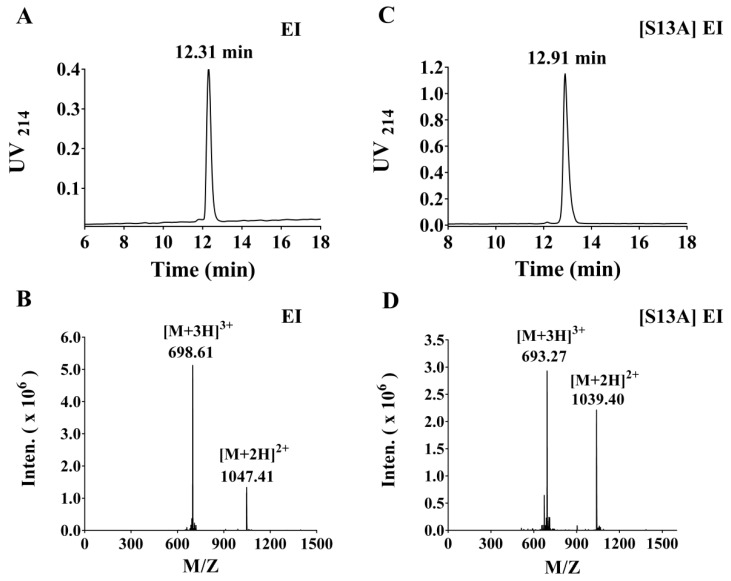
The HPLC and ESI–MS profiles of α-CTx EI and [S13A] EI. The peptide EI was purified to a single compound using a reversed-phase analytical Vydac C18 column, eluted over a linear gradient 10–45% buffer B for 20 min, where buffer A = 0.075% TFA, remainder H_2_O and buffer B = 0.050% TFA, 90% acetonitrile, remainder H_2_O. (**A**) HPLC chromatogram of fully oxidized and folded peptide EI. (**B**) A monoisotopic mass of 2092.82 Da (calculated 2092.84 Da) for EI was observed in the ESI–MS spectrum. (**C**) HPLC chromatogram of fully oxidized peptide [S13A] EI. (**D**) ESI–MS data for [S13A] EI with observed monoisotopic mass of 2076.80 Da (Calculated 2076.82 Da).

**Figure 2 toxins-11-00603-f002:**
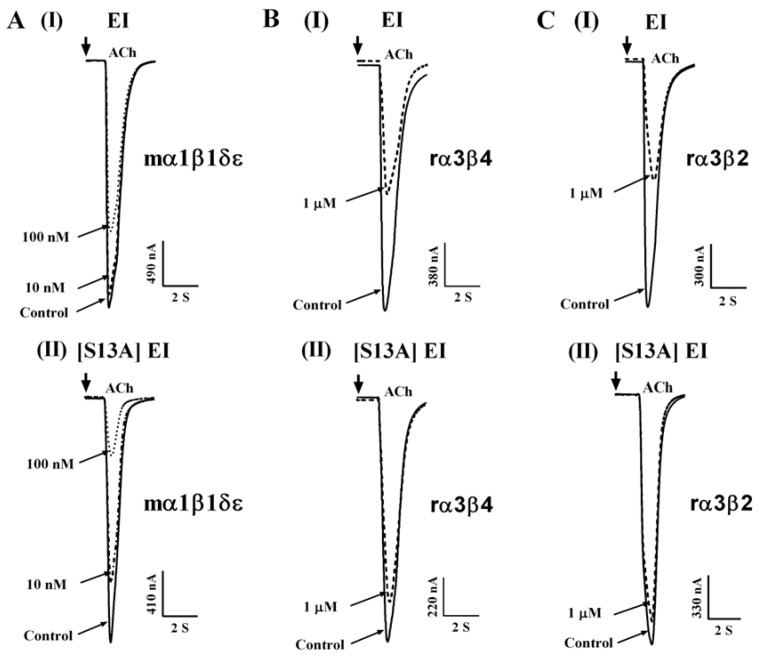
[S13A] EI inhibition of α1β1δε, α3β2, and α3β4 nicotinic acetylcholine receptors (nAChRs) compared with WT EI inhibition of these receptors. Cloned mouse α1β1δε (**A**), rat α3β4 (**B**), rat α3β2 (**C**) nAChR subtypes heterologously expressed in *Xenopus laevis* oocytes were recorded by TEVC. Superimposed traces representative of ACh-evoked current inhibition of α1β1δε (**A**), α3β4 (**B**), and α3β2 (**C**) nAChR subtypes by EI (I) and [S13A] EI (II). All data represent mean ± S.E.M, *n* = 3–5.

**Figure 3 toxins-11-00603-f003:**
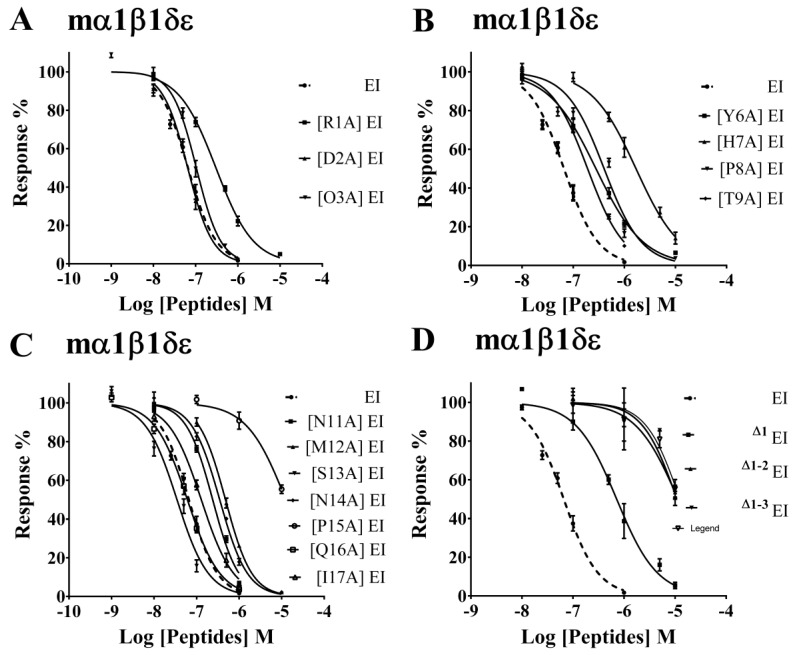
Effect of α-CTx EI and analogues at the mouse α1β1δε nAChR. (**A**) Concentration–response analysis for inhibition of mouse α1β1δε nAChR by Ala-substituted analogues in N-terminal “tail” amino acids. (**B**) Concentration–response curves for the inhibitory of mouse α1β1δε nAChR by EI analogues with Ala substitutions in the loop1 region. (**C**) The inhibition of mouse α1β1δε nAChR by EI analogues with Ala substitutions in the loop2 region was analyzed by concentration–response studies. (**D**) Concentration–response analysis for inhibition of mouse α1β1δε nAChR by N-terminally truncated analogues in α-CTx EI. All data represent mean ± S.E.M., *n* = 6–8.

**Figure 4 toxins-11-00603-f004:**
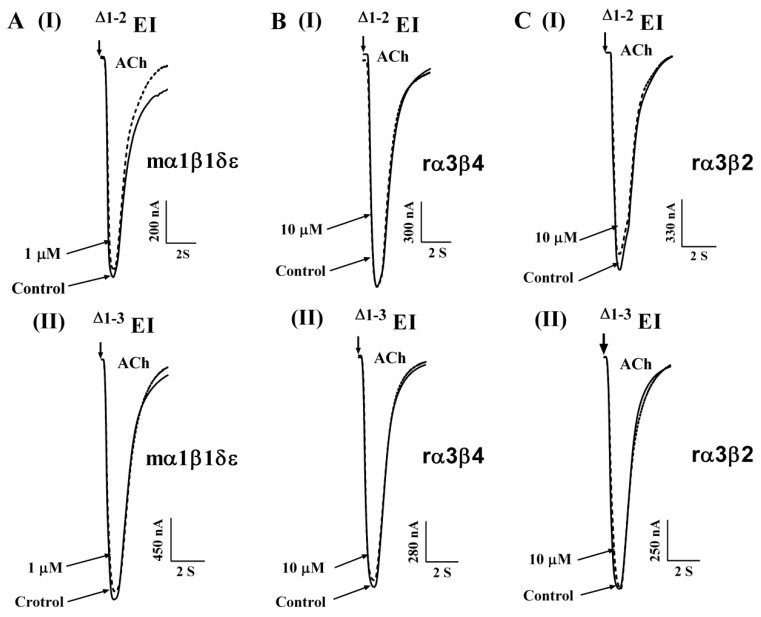
The effect on mouse α1β1δε expressed in *Xenopus laevis* oocytes by N-terminal truncated analogues. Mouse α1β1δε (**A**), rat α3β4 (**B**), rat α3β2 (**C**) nAChR subtypes expressed in *Xenopus* oocytes were activated by ACh. Superimposed traces representative of ACh-evoked current inhibition of α1β1δε (**A**), α3β4 (**B**), and α3β2 (**C**) nAChR subtypes by ^△1–2^ EI (I) and ^△1–3^ EI (II). All data represent mean ± S.E.M, *n* = 3–5.

**Figure 5 toxins-11-00603-f005:**
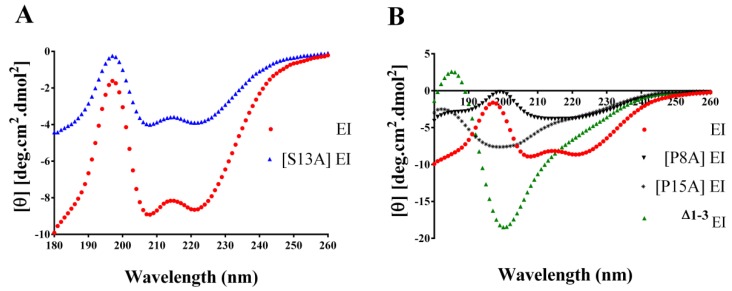
Characteristics of EI and its analogues. Circular dichroism (CD) spectra of Ala substitutions in the sequence of α-CTx EI compared with native globular EI. (**A**). It revealed that [S13A] EI mutant had similar spectra to globular EI. (**B**). [P8A] EI, [P15A] EI, and ^△1–3^ EI analogues do not display typically the α-helix characteristic and exhibit just a positive peak, indicating their secondary structures are disrupted.

**Table 1 toxins-11-00603-t001:** Sequences of EI and its analogues.

Peptide Number	Name	Sequences
1	EI	RDOCCYHPTCNMSNPQIC *
2	[R1A] EI	ADOCCYHPTCNMSNPQIC *
3	[D2A] EI	RAOCCYHPTCNMSNPQIC *
4	[O3A] EI	RDACCYHPTCNMSNPQIC *
5	[Y6A] EI	RDOCCAHPTCNMSNPQIC *
6	[H7A] EI	RDOCCYAPTCNMSNPQIC *
7	[P8A] EI	RDOCCYHATCNMSNPQIC *
8	[T9A] EI	RDOCCYHPACNMSNPQIC *
9	[N11A] EI	RDOCCYHPTCAMSNPQIC *
10	[M12A] EI	RDOCCYHPTCNASNPQIC *
11	[S13A] EI	RDOCCYHPTCNMANPQIC *
12	[N14A] EI	RDOCCYHPTCNMSAPQIC *
13	[P15A] EI	RDOCCYHPTCNMSNAQIC *
14	[Q16A] EI	RDOCCYHPTCNMSNPAIC *
15	[I17A] EI	RDOCCYHPTCNMSNPQAC *
16	^△^^1^ EI	DOCCYHPTCNMSNPQIC *
17	^△^^1–2^ EI	OCCYHPTCNMSNPQIC *
18	^△^^1–3^ EI	CCYHPTCNMSNPQIC *

The framework of disulfide-bond Cys are characterized in boldface and boxed. Disulfide connectivity of α-CTx EI and its analogues is between Cys1–Cys3 and Cys2–Cys4. Each substituted Alanine is labeled in bold and red. Asterisks denote a C-terminal amide. ^△1^ EI, ^△1–2^ EI, and ^△1–3^ EI indicate truncating the N-terminus in α-CTx EI sequentially by one residue, two residues, and three residues, respectively.

**Table 2 toxins-11-00603-t002:** The potencies of Ala-substituted α-CTx EI and its analogues on muscle and neuronal nAChRs expressed in *Xenopus laevis* oocytes.

	mα1β1δε nAChR			rα3β2 nAChR			rα3β4 nAChR		
Peptides	IC_50_ (nM)	nH	^a^ Ratio	IC_50_ (nM)	nH	^a^ Ratio	IC_50_ (nM)	nH	^a^ Ratio
1	65.9 (58.5–74.2)	1.3	1	7297 (3748–14,200)	0.8	1	6444 (5443–7628)	1.4	1
2	302 (265–346)	1.0	4.6	>10,000			>10,000		
3	64.8 (57.1–73.6)	1.5	1.0	>10,000			>10,000		
4	104 (91–119)	1.5	1.6	>10,000			>10,000		
5	278 (239–324)	0.9	4.2	~10,000			547 (435–687)	1.1	0.08
6	1688 (1407–2025)	1.0	25.6	>10,000			>10,000		
7	401 (338–477)	1.2	6.1	>10,000			>10,000		
8	191 (173–211)	1.2	2.9	65.3 (56.2–76.0)	1.3	0.009	603 (510–714)	1.3	0.09
9	239 (213–268)	1.4	3.6	>10,000			>10,000		
10	477 (441–517)	1.4	7.3	>10,000			>10,000		
11	34.6 (28.2–42.4)	1.2	0.5	>10,000			>10,000		
12	349 (315–386)	1.3	5.3	>10,000			>10,000		
13	12,510 (9311–16,810)	0.9	190	~10,000			>10,000		
14	60.6 (52.9–69.4)	1.1	0.9	~10,000			>10,000		
15	129 (110–151)	1.1	2.0	>10,000			>10,000		
16	716 (566–907)	1.1	11	>10,000			>10,000		
17	13,020 (10,300–16,440)	1.3	198	>10,000			>10,000		
18	12,190 (9394–15,810)	1.3	185	>10,000			>10,000		

^a^ indicates EI analogues in the ratio of IC_50_ values relative to EI. >1 indicates a decrease in potency, whereas <1 indicates an increase in potency. nH indicates Hill slope.

**Table 3 toxins-11-00603-t003:** Known α4/7-CTxs blocking various nAChR subtypes.

α-CTx	Sequences	Target	Reference
EI	RDOCCYHPTCNMSNPQIC *	muscle, α3β4, α3β2	[15]
PIA	RDPCCSNPVCTVHNPQIC *	α6/α3β2β3, α6/α3β4, α3β4, α3β2	[25]
GID	IRDγCCSNPACRVNNPHVC	α4β2, α3β2, α7	[24]
LoIa	EGCCSNPACRTNHPEVCD	α7, α3β4, muscle	[2]
Mr1.7	PECCTHPACHVSHPELC *	α3β2, α9α10	[26]
RegIIA	GCCSHPACNVNNPHIC *	α3β4, α3β2, α7	[23]

The framework of disulfide-bond Cys are characterized in boldface and boxed. O, hydroxyproline, and γ, γ-carboxyglutamic acid, and * indicates a C-terminal amide.

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
