# Peer review of "Identification of Crucial Residues in α-Conotoxin EI Inhibiting Muscle Nicotinic Acetylcholine Receptor"

_toxins, 2019, doi:10.3390/toxins11100603_

Round 1

Reviewer 1 Report

This paper, "Identification of Crucial Residues in a-Conotoxin EI Inhibiting Muscle Nicotinic Acetylcholine Receptor" reports alanine screening mutagenesis of a4/7-CTx EI in an effort to examine the interaction between EI and a1b1de nAChR.  The study is comprehensive and well-reported.  I offer the following recommendations for edits.

Page 2, line 61: sentence punctuation is required following analogues.

Page 2, Table 1: the table formatting is off-shifted between the first two columns and the third column.

Page 3, Line 85: The mass should be 2076.80 Da rather than 2070.80 Da as written.  It is correctly stated in the figure legend for Figure 1.

Table 2: In multiple places the activity for the d1-2 peptide mutant, number 17, is stated as being less active than either the d1 or the d1-3 (peptides 16 and 18).  Is there a statistically significant difference in the activity for the d1-2 and the d1-3, peptides 17 and 18?  In the discussion it is stated that neither peptide is active toward the a1b1de nAChR.  How many trials were run to obtain the data presented in Table 2? Is there a reason peptide 17 would lose more activity than peptide 18?  It seems that the N-terminal amino acids would systematically deteriorate in activity as amino acids are removed, but instead the second amino acid omission has a bigger impact on activity loss than the three amino acid omission.  Is there a theory for this finding?

Page 8, lines 169-170: It seems odd that the a-helix, which is dependent on the Pro-His for initiation, results in a random coil structure when the three N-terminal amino acids are removed.  The literature seems to support the notion that the Pro in combination with disulfide bonds are the dominate drivers for the a-helix scaffold for the a4/7-CTxs.  How do the authors explain the observation that the three N-terminal amino acids destroy the common scaffold for a4/7-CTxs?

Page 9-10, beginning line 234: A discussion of a-CTxs PIA, GID, and EI provides precedent for activity loss upon removal of N-terminal amino acids. A figure would complement this discussion well.  It was stated that NMR and X-ray structures are available for EI.  Images of the three peptides (a-CTxs PIA, GID, and EI) would help the reader to see the peptides being compared, the amino acids being modified, and correlate visual depictions of the peptides to the discussion being provided. 

Author Response

Dear editors and reviewers: Thank you for giving me the opportunity to revise and resubmit this manuscript (Manuscript ID: toxins-606587), these comments are all valuable and very helpful for revising and improving our paper, as well as the important guiding significance to our researches. All these comments are very valuable and helpful for revising and improving our paper, as well as guiding our researches. We have made revision to these comments after studied them carefully, which are marked in red. We have also revised our manuscript according to the comments. The revised version can be found in attached files, which we would like to submit for your kind consideration. The responds to the reviewer’s comments are as following:  Response to the reviewer’s comments: Reviewer 1 Comment 1: sentence punctuation is required following analogues. The table 1 formatting is off-shifted between the first two columns and the third column. The mass should be 2076.80 Da rather than 2070.80 Da as written. It is correctly stated in the figure legend for Figure 1. Response 1: Thanks for reviewers’ kind suggestions I have thoroughly checked my figures and tables in this time. Revision in introduction, figure 1 and Table1 (line 61, line 85 and figure 1 in page 2) thank you! Comment 2: In multiple places the activity for the d1-2 peptide mutant, number 17, is stated as being less active than either the d1 or the d1-3 (peptides 16 and 18). Is there a statistically significant difference in the activity for the d1-2 and the d1-3, peptides 17 and 18? In the discussion it is stated that neither peptide is active toward the α1β1δε nAChR. How many trials were run to obtain the data presented in Table 2? Is there a reason peptide 17 would lose more activity than peptide 18? It seems that the N-terminal amino acids would systematically deteriorate in activity as amino acids are removed, but instead the second amino acid omission has a bigger impact on activity loss than the three amino acid omission. Is there a theory for this finding? Response 2: Thanks for reviewers’ questions! In our trials, the IC50 values of EI analogues were obtained from at least 3 independent experiments. All data were obtained from 6 to 8 different Xenopus laevis oocytes. Then, these data were systematically analyzed by Graphpad software and the concentration-response curves for EI and analogues were fitted by nonlinear regression to the logistic equation, % response = 100/{1 + ([toxin]/IC50)nH}. The IC50 values of two EI analogues (peptide 17 and peptide 18) which inhibited α1β1δε nAChR were 13.0 μM and 12.2 μM, respectively. For conotoxins, if their IC50 values were greater than 10 μM, they are usually considered to be lacking potency for nAChR subtypes. Thus, two mutants (peptide 17 and peptide 18) all displayed a loss in antagonist activity for mouse α1β1δε nAChR, compared with wild-type EI. They have no significant difference in potency for α1β1δε nAChR. The CD spectroscopy results demonstrated that secondary structure of analogue △1-3 EI was disturbance. We speculated that analogue △1-3 EI secondary structure changes may have been involved in the loss of its activity. Similar results have been found in other studies, truncating the N terminus by residues 1-4 GID (isoleucine, arginine, aspartic acid, and γ-carboxyglutamic acid) in α-CTx GID also resulted in inactive for the main receptor α4β2 nAChR [Reference 27]. Therefore, we demonstrated that the triple amino acids “RDO” at the N-terminus play a crucial role in maintaining the potency of EI for mouse α1β1δε nAChR, and we also add these issues to the discussion section. Comment 3: Page 8, lines 169-170: It seems odd that the a-helix, which is dependent on the Pro-His for initiation, results in a random coil structure when the three N-terminal amino acids are removed. The literature seems to support the notion that the Pro in combination with disulfide bonds are the dominate drivers for the α-helix scaffold for the α4/7-CTxs. How do the authors explain the observation that the three N-terminal amino acids destroy the common scaffold for α4/7-CTxs? Response 3: Thanks! A conserved SHPA motif in loop1 region in α4/7-CTxs is common and responsible for receptor binding activity [Reference 11, 12]. The amino acids His and Pro are thought to confer structural rigidity to the α-helical structure of α-CTxs [Reference 27]. The CD spectroscopy result demonstrated that peptide EI, with a positive ellipticity at 195 nm (λ) and two negative ellipticities at 208 nm (λ) and 222 nm (λ), respectively. This indicated that EI were predominantly indicative of α-helical. Replacement of Pro-15 with Ala and the truncated analogue △1-3 EI resulted in a remarkable change in spectral characteristic and the α-helical disappear. Comment 4: Discussion of a-CTxs PIA, GID, and EI provides precedent for activity loss upon removal of N-terminal amino acids. A figure would complement this discussion well. It was stated that NMR and X-ray structures are available for EI. Images of the three peptides (a-CTxs PIA, GID, and EI) would help the reader to see the peptides being compared, the amino acids being modified, and correlate visual depictions of the peptides to the discussion being provided. Response 4: I have added the figure about comparison of α- CTxs PIA, GID, and EI into supporting information. All three peptides all have extra N-terminal amino acids tails. In our discussion section we add some contents (line 248, 252) based on three peptide structure.

Reviewer 2 Report

In the present work, authors have identified the important residues of the alpha-conotoxin EI, which contribute to inhibiting effect of the toxin on muscle-specific nAChRs. The paper is well organized and illustrated. I have only minor questions.

Minor comments:

Could you check the ability of the mutant toxins to modulate nAChR at the mice neuromuscular junctions? It may be important, due to organization of the receptors in cluster in the end plate, and only transient increase in endogenous AChR level during synaptic exocytosis. So, the conditions in Xenopus and in NMJs are different.

As experiments were performed at RT, could you please mention a possible influence of physiological temperature on effects of the mutant toxins?

In line 5-6: Known that deficits of nAChRs contribute to development of these disorders. I do not sure about using “recently”.

Line 7: Please indicate the subunit compositions of muscle-specific nAchRs

Line 11: I suggest add “voltage-clamp recording”. Also, this method can provide information about currents through membrane channels under different condition; but it cannot indicate “the key residues”. Please re-phrase.

Line 42-43: Please clarify subunit compositions of neuronal nAchRs.

In Discussion:

May action of the toxin be dependent on lipid environment (lipid raft, cholesterol)? Please, If it possible, could you extend the discussion section with this point?

Author Response

Dear editors and reviewers:
Thank you for giving me the opportunity to revise and resubmit this manuscript (Manuscript ID: toxins-606587), these comments are all valuable and very helpful for revising and improving our paper, as well as the important guiding significance to our researches. All these comments are very valuable and helpful for revising and improving our paper, as well as guiding our researches. We have made revision to these comments after studied them carefully, which are marked in red. We have also revised our manuscript according to the comments. The revised version can be found in attached files, which we would like to submit for your kind consideration.

The responds to the reviewer’s comments are as following: 

Response to the reviewer’s comments:

Reviewer 2

Comment 1: Could you check the ability of the mutant toxins to modulate nAChR at the mice neuromuscular junctions? It may be important, due to organization of the receptors in cluster in the end plate, and only transient increase in endogenous AChR level during synaptic exocytosis. So, the conditions in Xenopus and in NMJs are different.

Response1:Thanks for reviewers’ question and advice., The muscle-type receptors are mainly distributed at the skeletal neuromuscular junction and muscle-type nAChRs have many different compositions such as α1β1δε, α1β1δγ, α1β1δ, α1β1γ[Reference5,15]. The α1β1δε muscle-type nAChR is one of the most important subtype, and a target widely application in pharmacology, pathology and drug development. The main purpose of our study is to use Ala-scan mutagenesis method to identify the key amino acid residue and find novel EI analogue which shows high selectivity for α1β1δε. It may also be important in remodeling of some novel ligands for inhibiting muscle-type nAChRs. Currently, the mutants have not been used in animal experiments. Although the α1β1δε model expressed in frog egg cells is quite different from in NMJs, but the use of the Xenopus oocyte translation system for transient expression has proved to be a critical tool in the molecular characterization of ion channels and receptor [Reference6]. This system has been used to confirm the identify of cDNAs encoding voltage-dependent ion channels and numerous receptors, such as K+ channel, Na+ channel and nAChR.

Comment 2: As experiments were performed at RT, could you please mention a possible influence of physiological temperature on effects of the mutant toxins?

Response 2: all experiments were performed in vitro. Because, Xenopus laevis oocytes is a model species that is used to neuroscience field, such as ion channels and receptors. Inhibition of neuronal nicotinic acetylcholine receptor subtypes by alpha-Conotoxin GID reveals the critical residues for activity at α4β2 nAChR [Reference 27] and α-Conotoxin PIA selectively inhibited α6 nAChRs [Reference 25]. In addition, all Xenopus laevis oocytes were cultured in RT. Therefore, we do not physiological researches and we will also do experiment in vivo such as cells and mouse next. Thanks!

Comment 3: In line 5-6: Known that deficits of nAChRs contribute to development of these disorders. I do not sure about using “recently”. Line 7: Please indicate the subunit compositions of muscle-specific nAChRs, I suggest add “voltage-clamp recording”. Also, this method can provide information about currents through membrane channels under different condition; but it cannot indicate “the key residues”. Please re-phrase. Line 42-43: Please clarify subunit compositions of neuronal nAChRs.

Response 3: it is very kind for reviewers’ suggestions! I have completely checked and modified these errors. Revisions were in line 4, 7, 9 and 27 in Abstract section and line 46 in introduction section.

Comment 4: In Discussion: May action of the toxin be dependent on lipid environment (lipid raft, cholesterol)? Please, If it possible, could you extend the discussion section with this point?

Response 4: nAChR is divided into three domains: an N-terminal extracellular (LBD), A membrane-spanning pore and intracellular domain. We mainly study the relationship between ligands and receptors and rationally design some analogues. these analogues interact with nAChR in N-terminal extracellular. Therefore, in this report, we do not add lipid environment into discussion section, thanks!
